# Recent Advances in the Development of Adenovirus-Vectored Vaccines for Parasitic Infections

**DOI:** 10.3390/ph16030334

**Published:** 2023-02-22

**Authors:** Cal Koger-Pease, Dilhan J. Perera, Momar Ndao

**Affiliations:** 1Division of Experimental Medicine, McGill University, Montreal, QC H3A 3J1, Canada; 2Infectious Diseases and Immunity in Global Health (IDIGH) Program, Research Institute of the McGill University Health Centre, Montreal, QC H4A 3J1, Canada; 3Department of Microbiology and Immunology, McGill University, Montreal, QC H3A 0G4, Canada; 4National Reference Centre for Parasitology, Research Institute of McGill University Health Centre, Montreal, QC H4A 3J1, Canada

**Keywords:** adenovirus vaccines, parasites, protozoa, helminth, vaccine vector

## Abstract

Vaccines against parasites have lagged centuries behind those against viral and bacterial infections, despite the devastating morbidity and widespread effects of parasitic diseases across the globe. One of the greatest hurdles to parasite vaccine development has been the lack of vaccine strategies able to elicit the complex and multifaceted immune responses needed to abrogate parasitic persistence. Viral vectors, especially adenovirus (AdV) vectors, have emerged as a potential solution for complex disease targets, including HIV, tuberculosis, and parasitic diseases, to name a few. AdVs are highly immunogenic and are uniquely able to drive CD8+ T cell responses, which are known to be correlates of immunity in infections with most protozoan and some helminthic parasites. This review presents recent developments in AdV-vectored vaccines targeting five major human parasitic diseases: malaria, Chagas disease, schistosomiasis, leishmaniasis, and toxoplasmosis. Many AdV-vectored vaccines have been developed for these diseases, utilizing a wide variety of vectors, antigens, and modes of delivery. AdV-vectored vaccines are a promising approach for the historically challenging target of human parasitic diseases.

## 1. Introduction

Vaccination as we know it today dates to 1798 when Edward Jenner first developed a smallpox vaccine using the milder but still protective cowpox virus [1]. The idea of inoculation, however, is significantly older. In the 18th century, a practice known as variolation was first introduced to Europe, although it was used in non-Western cultures as early as 1000 B.C. [1]. Variolation involves inoculating a healthy individual with the infectious material from an infected person’s “pox” by blowing dried smallpox scabs into their nostrils. Inoculated individuals would generally contract a mild form of the disease and recover, in the process becoming immune to smallpox [1,2]. Compared to a natural death rate of 14%, only 2% of individuals who were variolated died from smallpox [1]. However, this technique was not without risk; some did die from variolation, and as it caused true infections, inoculated individuals could spread disease [1]. A similar inoculation against the parasite *Leishmania,* called Leishmanization, has also been used for thousands of years [3].

Since Jenner’s first vaccine, numerous others have been developed against devastating infectious diseases such as rabies, cholera, tuberculosis, yellow fever, influenza, mumps, and hepatitis B, to name a few [4]. Many traditional vaccines use either a live-attenuated or whole-killed pathogen approach. Live-attenuated vaccines, such as Jenner’s cowpox vaccine, involve the use of a weaker pathogen, whether naturally occurring or laboratory-attenuated [5]. Such vaccines closely mimic natural infection and thus natural immunity, though there is a risk that vaccination may cause disease or that attenuated pathogens may revert to more virulent forms [5]. Whole inactivated or killed pathogen vaccines solve this safety issue; however, immunogenicity is often severely reduced [5].

Many recently developed vaccines have taken a subunit approach. Subunit vaccines take specific immunogenic antigens from pathogens and use those to “educate” the immune system to recognize the entire infectious agent. These vaccines can involve direct delivery of antigen(s) as recombinant proteins, or they can provide DNA or RNA sequences which are then transcribed and/or translated within the vaccinated individual [4]. One method of delivering nucleic acid-based vaccines is the use of a viral vector. The first viral vector vaccine was developed in the 1980s using a recombinant vaccinia virus expressing the surface antigen of hepatitis B virus (HBV), providing protection against HBV infection [6]. While promising, this vaccine and other similar ones have not been approved in humans, in large part due to the use of replication-competent vectors, which raise serious safety concerns [7]. Alternatively, viral vector vaccines can be replication-incompetent, bolstering their safety profile [8]. One of the most compelling rationales for the use of viral-vectored vaccines is their ability to elicit strong humoral *and* cell-mediated immune responses [8]. Many different viral vectors have been used in the context of vaccine development, including adenoviruses (AdVs), adeno-associated viruses, lentiviruses, and poxviruses [8]. Currently, AdV vectors are the most widely used viral vectors in vaccine development [9]. As a vaccine vector, AdVs have numerous advantages, including broad cell and tissue tropism, a well-characterized safety profile, high immunogenicity (especially the induction of CD8+ T cell responses), and significant transgene incorporation and expression ability [8]. Furthermore, during the COVID-19 pandemic, multiple AdV-vectored vaccines were rapidly developed and deployed, demonstrating the applicability of this vaccination approach [8,9,10]. The most widely distributed AdV-vectored COVID-19 vaccines are the AstraZeneca/Oxford (Vaxzevria) and Janssen/Johnson&Johnson (JCOVDEN) vaccines. The World Health Organization (WHO) estimates the efficacy against symptomatic disease of these two vaccines as 72% [11] and 94% [12], respectively; however, data are lacking on their efficacy against more recent variants. Both vaccines have been associated with very rare events of thrombosis with thrombocytopenia syndrome (TTS) but are still generally considered safe for adults over 18 years of age [11,12].

Nearly all vaccines against infectious diseases currently approved for human use target viral or bacterial infections. Traditional vaccine approaches have generally been most successful for diseases where immune protection is dependent on antibody responses [13]. There is a notable lack of vaccines that effectively elicit multifaceted immunity for complex pathogens such as parasites. Humans have been interacting and co-evolving with parasites for the entire span of our evolution [14,15], meaning that human-parasite interactions are incredibly convoluted, and both natural and vaccine-derived immunity is often limited [16,17]. Thus, viral-vectored vaccines have emerged as a potential solution to the paucity of efficacious parasite vaccines. Multiple viral vectors have been used to target parasites, most notably AdVs and poxviruses, both of which are promising in this context. AdVs have some subtle advantages, such as a longer period of antigen expression and a higher ratio of transgene expression to viral vector antigen expression [17]. This review presents recent developments in AdV-vectored vaccines against human parasitic infections, with a focus on five major parasitic diseases: malaria, Chagas disease, schistosomiasis, leishmaniasis, and toxoplasmosis.

## 2. Parasite Vaccines

Pathogenic human parasites can be classified as protozoa (single-celled eukaryotes) or helminths (worms). Malaria, caused by protozoan parasites of the *Plasmodium* genus, is by far the most devastating and deadly parasitic disease [18]. Many different parasitic diseases, however, contribute significantly to the overall global burden, such as schistosomiasis, leishmaniasis, echinococcosis, toxoplasmosis, human African trypanosomiasis, Chagas disease, and lymphatic filariasis, among others [19].

In 2021, the WHO recommended the widespread use of RTS,S/AS01 (RTS,S), a recombinant protein-based malaria vaccine, for children living in moderate to high *Plasmodium falciparum* transmission areas [20], officially marking the first-ever broad administration of a vaccine against malaria or any parasitic disease for that matter. The RTS,S vaccine has been in development for over 30 years, and trials and rollout have involved the cooperation and funding of industry, academia, and governments [20]. RTS,S is created by the co-expression in yeast of a fusion gene of repeat regions and C-terminal fragments of *P. falciparum* circumsporozoite protein (CSP) and the N-terminal region of hepatitis B surface antigen (HBsAg) together with an additional unmodified HBsAg expression cassette. The result is a lipoprotein particle that resembles HBsAg but expresses CSP fragments on the outside of the particle [21].

Although this vaccine is ground-breaking, there is still significant room for improvement. Phase III clinical trials of RTS,S showed 63% and 74% protection against clinical malaria in infants and children, respectively, at a few weeks post-vaccination, but by 5 years, protection dropped to only 1% and 9%, respectively [22]. The most recent estimates of the pilot rollout of RTS,S showed an overall 30% protection against severe malaria [23]. Recently, a new take on the RTS,S vaccine strategy has shown even greater protection in a phase I/IIb clinical trial in children in Burkina Faso [24]. This vaccine, called R21, eliminates the need for the second unmodified HBsAg component, creating a similar particle with a significantly higher ratio of CSP to HbsAg expression [25]. Because the protection generated by RTS,S is thought to be mediated by antibody responses to CSP, it follows that increased expression of CSP (and decreased expression of HbsAg) would increase the protective antigen-specific immune response [25]. In fact, the R21 vaccine was shown to induce significant anti-CSP antibody titers and very limited anti-HbsAg titers in mice when administered with a variety of adjuvants [25]. Moreover, R21 appears to be an improvement on RTS,S in humans, as it boasts an efficacy of up to 80% against clinical malaria after 1 year. Still, follow-up studies will be needed to assess long-term efficacy [24].

Other than RTS,S, there are no licensed vaccines for parasitic diseases. However, there has been progress towards vaccines against both protozoan (reviewed in [26,27]) and helminthic (reviewed in [28,29]) parasites. Additionally, there are multiple vaccines against zoonotic parasites licensed for animals (e.g., *Leishmania* and *Toxoplasma*), suggesting that human anti-parasite vaccines may be a reality in the near future [26]. AdV-vectored vaccines have been investigated as an excellent option for parasites, given their ability to elicit the multifaceted immune responses necessary for protection against parasites.

## 3. Adenoviruses as a Vaccine Vector

Adenoviruses are non-enveloped, double-stranded DNA (dsDNA) viruses with genomes ranging from 25–48 kb [30]. There are many species of AdVs that infect a broad range of vertebrate hosts, including mammals, reptiles, birds, and fish [30]. Of the human AdVs, there are 51 known serotypes classified into six subgroups (A-F) [31]. Natural AdV infection is typically mild, most often affecting the respiratory or gastrointestinal tract, yet AdVs are highly immunogenic, making them good candidates for vaccine vectors [31,32].

AdV vectors can either be replication-competent or replication-incompetent. Although both strategies have been used, replication-incompetent vectors are preferred because they avoid the risks associated with the use of a replicating virus. Replication-incompetent AdVs lack the E1 gene, which is necessary for viral replication. Deletion of E1 has the additional benefit of providing more space in the vector genome for the insertion of a transgene. The E3 gene is also typically deleted because E3 proteins have immunosuppressive effects, such as limiting major histocompatibility complex (MHC) class I expression and blocking TNF signaling pathways, which would limit the AdV vector’s immunogenicity [31]. Without the E1 and E3 regions, an AdV vector can accommodate up to 7.5 kb of foreign DNA [31].

### 3.1. Immune Responses to AdV Vectors

The immune reaction to AdVs is swift, beginning with powerful innate responses, as depicted in Figure 1. Within a matter of hours, AdV infection causes innate immune cells, notably dendritic cells (DCs) and macrophages, to begin the production of pro-inflammatory cytokines such as IL-6, IL-12, and TNF-α [33]. Soon after, the adaptive immune response begins, characterized by both humoral and cell-mediated immunity. When the viral vector infects a host cell, it produces the encoded antigen in the cytoplasm of the infected cell. Epitopes from this antigen are then expressed on the cell’s MHC class I, priming CD8+ T cell responses. The antigen and other immunogenic debris are also released upon infected cell death and then taken up by other macrophages. This allows for antigen epitope presentation on MHC class II molecules via exogenous antigen processing pathways, in turn priming CD4+ T cell responses [34]. AdV vectors have consistently been shown to induce potent transgene-specific CD8+ T cell responses and antibody titers, even more so than other viral vectors [31].

### 3.2. Limitations of AdV Vectors: Concerns and Considerations

One of the major concerns with the use of AdV vectors as vaccines is that pre-existing immunity to the virus in the general population will limit vaccine efficacy. The most well-studied AdV serotype for use in gene therapy and vaccines is human adenovirus serotype 5 (HAdV-5). While HAdV-5 has a promising safety and immunogenicity profile as a vaccine, natural infection with this serotype is extremely common; up to 90% seroprevalence has been documented in some parts of the world [35]. Such widespread seroprevalence is concerning given that previously infected individuals will have pre-existing immune memory to the virus, which could limit the ability of the vector to effectively deliver the transgene and spark immune responses [8,36]. Much of this concern is based on the STEP HIV vaccine trial, which found an association between high pre-existing antibody titers against HAdV-5 and low responses to the vaccine-encoded HIV transgenes [36]. That said, the mechanisms of this association are not fully understood, and whether pre-existing immunity precludes the successful use of HAdV-5-vectored vaccines in humans has not been conclusively demonstrated [8,31].

While HAdV-5-based vaccines continue to be developed and tested, many researchers have begun investigating other serotypes and modification techniques in parallel to overcome any deleterious effects of pre-existing immunity [7,8,36]. One of the most common techniques is using a less seroprevalent human AdV, such as HAdV-26, 35, or 48, or nonhuman AdVs, especially chimpanzee AdVs [7,8,36]. Another novel strategy is the “capsid integration” technique. Instead of incorporating a transgene into the viral genome, the antigen is expressed on the viral capsid proteins. This alters the “appearance” of the capsid, limiting the effect of pre-existing immunity. Additionally, antigens expressed in this way will be primarily recognized by exogenous processing pathways, which change the dynamics of the immune response. Capsid modifications can be used in combination with transgene approaches as well, broadening the immune response [37].

During the COVID-19 pandemic, multiple AdV-vectored vaccines were developed and approved in various countries, including the widely distributed Vaxzevria and JCOVDEN vaccines. Through wide-scale administration of Vaxzevria and JOVDEN, evidence was found that AdV-vectored vaccines may cause rare events of TTS, which, when associated with vaccination and the presence of antibodies to platelet factor 4 (PF4), is termed vaccine-induced thrombotic thrombocytopenia (VITT) [38,39]. The risk of TTS and VITT has been somewhat challenging to quantify due to major differences in reporting and diagnostic abilities in various countries, but estimates place the rate of TTS as 3.2 to 16.1 cases per million doses for Vaxzevria and 1.7 to 3.7 cases per million doses of JCOVDEN, while no cases have been associated with the mRNA vaccines [39]. The exact mechanism of VITT is not fully understood, but the working model suggests that the AdV capsid proteins form a complex with PF4, leading to the activation of B cells specific to PF4, which in turn leads to the production of autoantibodies against PF4 and a prothrombotic cascade [38,39]. VITT is so rare that it was not identified in phase III clinical trials and only became clear after the broad worldwide distribution of the vaccines. Nevertheless, such severe adverse reactions need to be investigated further and considered in the context of any AdV-vectored vaccine used in the future. It is possible that capsid modification strategies could reduce the risk of complex formation between the AdV vector and PF4. It appears that the interaction between the vector and PF4 depends on the electronegative residues in the AdV capsid, which could be modified to prevent binding. Once the exact binding location and mechanism are fully elucidated, even more specific modifications can be designed to reduce the risk of VITT and potentially also limit the effect of pre-existing antibodies at the same time [38].

### 3.3. AdV Vaccines Approved and in Development

Many AdV-vectored vaccines have been used in the COVID-19 pandemic, including the aforementioned Vaxzevria and JCOVDEN vaccines, as well as the Serum Institute of India Covishield, CanSino’s Ad5-nCoV, and Russian Sputnik V vaccines. These vaccines use a variety of AdV serotypes, including HAdV-5, HAdV-26, and ChAdOx1 (a chimpanzee serotype), all expressing the SARS-CoV-2 spike protein [10,40]. The rapid development and deployment of these vaccines were made possible by the significant body of research into AdV vectors for several diseases. Some notable clinical trials using AdV-vectored vaccines are summarized in Table 1. A broad range of diseases has been targeted with AdV-vectored vaccines, including HIV, Ebola, tuberculosis, and numerous cancers. AdV-vectored vaccines have shown significant potential in preclinical and clinical development, and many new vaccines are actively being investigated today.

AdV-vectored vaccine development for parasites has primarily focused on malaria and other protozoa. Given that one of the major benefits of AdV vectors is their ability to elicit CD8+ T cell responses, intracellular pathogens, such as protozoan parasites, are a logical target. That said, AdV-vectored vaccines have demonstrated protection in helminth infection as well, especially against *Schistosoma* species [41,42,43,44,45].

## 4. Adenovirus-Vectored Vaccines against Malaria

As briefly mentioned, malaria is a life-threatening febrile disease caused by protozoan parasites of the *Plasmodium* genus, most commonly *P. falciparum* and *P. vivax.* In 2021, there were an estimated 247 million cases and 619,000 deaths worldwide due to malaria, with the majority of those deaths occurring in children under five years of age. The deadliest malaria-causing parasite, *P. falciparum*, is the most common in Africa, while *P. vivax* is more common elsewhere in the world. Malaria is transmitted through the bite of female *Anopheles* mosquitos [46,47]. Typical symptoms of uncomplicated or non-severe malaria include fever, chills, sweating, headache, nausea, and vomiting. Complicated or severe malaria includes cerebral malaria, characterized by neurologic problems such as seizures, loss of consciousness, and coma; severe anemia; acute respiratory distress syndrome; low blood pressure; and other conditions [47].

The course of infection with *Plasmodium* can be broadly split into two phases: the pre-erythrocytic or liver stages and the erythrocytic stages. Pre-erythrocytic immunity is largely studied in the context of radiation-attenuated sporozoite (RAS) vaccines, which are still considered the “gold standard” vaccination approach for this parasite [48,49]. Vaccination with RAS in mice, non-human primates (NHP), and humans has been shown to provide sterile immunity against *Plasmodium* infection [49,50]. Natural infection, by contrast, provides protection from the erythrocytic stage (during which clinical disease occurs). However, this protection develops over many years of repeated infection and does not provide sterile immunity [49].

Malaria infection begins upon injection of the sporozoite form of the parasite from mosquitoes into the skin. Next, the sporozoite travels to the liver. The parasite must transit through the dermis, into the bloodstream, across the sinusoidal barrier of the liver, and finally into hepatocytes [49,51]. Inside the hepatocyte, the sporozoite replicates and develops into merozoites within a parasitic vacuole. The vacuole then bursts, and the mature merozoites flood the hepatocyte cytoplasm. Merozoites then bud from the hepatocyte in merosomes, which enter the blood stream for erythrocytic invasion [52]. The pre-erythrocytic stages are considered “clinically silent”; however, the host immune response is anything but. In mouse models, CD8+ T cells, IFN-γ, IL-12, inducible nitric oxide synthase (iNOS), and natural killer (NK) cells have all been identified as correlates of protection in RAS vaccination, suggesting that these immune effectors are important during the pre-erythrocytic stage. In addition to cell-mediated immunity, antibodies developed against pre-erythrocytic stage antigens, such as CSP, can confer protection when present in high titers [49].

Once the *Plasmodium* merozoites have been released into the blood stream, the erythrocytic cycle begins. This cycle consists of intraerythrocytic asexual reproduction, rupture of the erythrocyte and release of daughter merozoites, and reinfection of new erythrocytes by those daughter merozoites, thus amplifying the infection [53]. In contrast to the pre-erythrocytic stage, the erythrocytic stage results in clinical disease; the symptoms of malaria are due to immune activation by merozoites and other products released into the blood stream upon erythrocytes bursting [49]. Innate immune responses in the erythrocytic stages include toll-like receptor (TLR) 2 and 9 sensing and MyD88 signaling pathways; production of IFN-γ, TNF-α and IL-1; and the recruitment of macrophages [49]. In this stage, both cell-mediated and humoral immunity are important.

Although the RTS,S vaccine has provided an important tool in the anti-malaria arsenal, there is a significant need for more effective and long-lasting vaccines. Viral-vectored vaccines are a well-studied option for malaria prevention, and many different viral vectors and antigens have been tested. The most advanced of these involve a heterologous prime-boost of chimpanzee adenovirus 63 (ChAd63) prime followed by a modified vaccinia Ankara (MVA) boost, each expressing one or more *Plasmodium* antigens.

In the past ten years alone, more than 20 different ChAd63/MVA vaccines or vaccine regimens have been developed and tested by researchers at the Jenner Institute of Oxford University [25,54,55,56,57,58,59,60,61,62,63,64,65,66,67,68,69,70,71,72,73,74,75,76,77,78,79,80,81,82,83,84,85,86,87,88,89,90], the same institute responsible for the R21 vaccine [24,25]. The most advanced and successful of these vaccines encodes the *P. falciparum* Thrombospondin-Related Adhesion Protein (TRAP) antigen fused to a string of additional malaria epitopes (ME). TRAP is a vital pre-erythrocytic antigen that the parasite uses for hepatocyte invasion [48]. ChAd63/MVA ME-TRAP has demonstrated a consistently favorable safety profile across different ages and populations in phase I and II clinical trials, even when combined with other vaccines and/or antigens [54,55,56,57,58,59,60,61,62,63,66,71]. However, efficacy has been more varied, especially between different populations and areas with divergent transmission dynamics. A phase I/IIa trial in healthy, malaria naïve UK adults showed 21.4% sterile immunity and a mean 2.8 day increase in time to patent blood-stage parasitemia after controlled human malaria infection (CHMI) [64]. In an additional phase I/IIa trial in UK adults, 18% of individuals vaccinated with ChAd63/MVA ME-TRAP achieved sterile protection against CHMI, and 33% of vaccinated individuals had a delay in the time of treatment [55]. In a phase IIb trail in Kenyan adults, this vaccine showed a higher efficacy of 67% (measured as the percent of patients remaining PCR-negative throughout their 56 day follow-up) or 82% (measured as the percent of patients maintaining low levels of parasitemia (≤10 parasites/mL)) [54]. However, in another trial in Senegal, this vaccine provided no significant protection, with an efficacy of 8%. The authors present multiple rationales for these differential results. For one, rates of malaria infection during the trial period in Senegal were lower than predicted, limiting the statistical power of the study. Secondly, malaria transmission is temporally different between Kenya and Senegal: in Kenya, transmission has peak seasons but is essentially year-round, while in Senegal, most (if not all) transmission occurs within a brief three-month wet season. The results of this trial highlight the importance of considering such transmission dynamics in the design of field studies. Lower transmission rates in the Senegalese trial cohort likely correspond with a lower baseline immunity, which in turn affects the vaccine efficacy. In the Kenyan cohort, even the initial dose of the vaccine may be more akin to a boost as the vaccinee already has a significant degree of protection from long-term continuous *Plasmodium* exposure. This hypothesis also explains why vaccine efficacy was increased in the Kenyan cohort over the malaria naïve UK cohort. Furthermore, TRAP-specific IFN-γ ELIspot results in the Senegalese population were notably lower than those in the Kenyan study, suggesting that the differences between the two populations affect the immunogenicity as well as the efficacy of the vaccine [56]. Similar to the results observed out of Senegal, a phase IIb trial of ChAd63/MVA ME-TRAP in children ages 5–17 months in Burkina Faso showed no significant vaccine efficacy against clinical malaria (13.8% protection) [61]. Taken together, the results of these clinical trials indicate that while ChAd63/MVA ME-TRAP showed promise in Kenya, it is not able to provide the protection needed for a successful malaria vaccine when used alone in varying locations. To that end, combinations of ChAd63/MVA ME-TRAP with other antigens and vaccines have been tested, including the RTS,S [57,62,87] and R21 vaccines [25]. The combination of ChAd63/MVA ME-TRAP and RTS,S did not provide any greater protection than RTS,S alone in clinical trials [57,62]. Combining R21 (adjuvanted with MF59) with ChAd63/MVA ME-TRAP, on the other hand, does provide increased protection for BALB/c mice. Co-administration of these vaccines protected 62.5% of mice from infection, which was a statistically significant increase in protection over each vaccine regimen alone [25]. A phase I/IIa CHMI study in healthy UK adults using this combined regimen has been completed; however, efficacy results are not yet published (NCT02905019).

ChAd63/MVA vaccines have also been developed against *P. vivax* malaria. One such vaccine targets the *P. vivax* Duffy binding protein (PvDBP), which is necessary for parasite entry into host red blood cells [74,85]. Preclinical studies in mice and rabbits demonstrated that ChAd63/MVA PvDBP is immunogenic and that vaccine-induced antibodies can recognize *P. vivax* merozoites in vitro [74]. Based on this preclinical data, the vaccine was tested in healthy UK adults, where safety and immunogenicity were confirmed in humans [85]. Currently, a stage IIa trial is underway to determine protective efficacy against CHMI (NCT04009096).

Several other AdV-vectored vaccines have been tested against malaria in preclinical models, but none have yet surpassed the protection achieved with ChAd63/MVA regimens. Some notable vaccines include a heterologous prime/boost protocol using HAdV-5 and adeno-associated virus (AAV) 8 or 1 vectors expressing *P. falciparum* CSP (PfCSP) [91], a capsid integration approach with PfCSP epitopes integrated into the HAdV-5 hexon and core protein VII [92], and a transmission-blocking vaccine using HAdV-5 expressing Pfs25 (a sexual stage protein) [93]. The complexity of the *Plasmodium* life cycle and host-parasite interactions means that an ideal vaccine may need to target multiple life stages and several antigens; thus, the continued investigation of novel malaria vaccines is essential. Current vaccines can also be tested for complementary and synergistic effects, further boosting the anti-malaria arsenal.

## 5. Adenovirus-Vectored Vaccines against Chagas Disease

Chagas disease, or American trypanosomiasis, is caused by the protozoan parasite *Trypanosoma cruzi*. The disease is primarily transmitted by the feces and urine of triatomine bugs but can also be transmitted congenitally and through organ transplants or blood transfusions from infected donors. Chagas disease is endemic in continental Latin America, but human migration has resulted in infected individuals around the world. An estimated 6–7 million people are infected worldwide. There are acute and chronic phases of clinical Chagas disease, although many infected individuals will remain asymptomatic throughout their life. Symptoms of the acute disease last 1 to 2 months post-infection and include fever, headache, nausea/vomiting, diarrhea, swollen lymph nodes, muscle or abdominal pain, and difficulty breathing. A rare but characteristic sign of acute infection is a swelling at the site of infection, called a chagoma, or Romaña’s sign in the case of infection around the eye mucosa. Up to three decades after initial infection, 30% of infected individuals will develop cardiac conditions such as cardiomyopathy, heart rhythm abnormalities, apical aneurysm, and/or sudden death/heart failure. Another 10% will develop gastrointestinal complications, typically enlargement of the esophagus or colon [94,95].

*T. cruzi* is an intracellular pathogen with broad cell and tissue tropism [96,97,98,99,100]. As with other intracellular pathogens, the body’s first reaction to infection is an innate immune response characterized by the production of IFN-γ [100]. *T. cruzi* sensing appears to occur through TLR-2 and TLR-9, although other TLRs and pattern recognition receptors (PRRs) may be important as well [100,101]. IFN-sensing genes (ISGs) are upregulated in the early immune response, and innate immune cells such as NK cells and macrophages are recruited to the site of infection [102]. The production of reactive oxygen species (ROS), reactive nitrogen species (RNS), and nitric oxide (NO) by innate immune cells is important for parasite killing. However, *T. cruzi* has evolved a strong antioxidant system, meaning high levels of these toxic mediators are necessary for parasiticidal activity, contributing to off-target tissue damage and inflammation in the host [100,103]. Interestingly, the CD8+ T cell response to *T. cruzi* is significantly delayed compared to other intracellular pathogens, possibly because of insufficient TLR activation. This hypothesis is supported by the observation that the addition of TLR agonists during experimental *T. cruzi* infection significantly accelerates parasite-specific CD8+ T cell expansion, more so than simply increasing the infectious dose [104]. Because of this delayed response, the parasite is often able to establish itself in the host before immune recognition [104]. Once activated, CD8+ T cell responses become very important for the killing of infected cells via cytotoxic lymphocyte (CTL) effector pathways and amplification of innate immune cell production of ROS/RNS/NO via secretion of IFN-γ [100]. In order to prevent severe immunopathology, immune-dampening mechanisms are activated later in infection, especially the production of anti-inflammatory cytokine IL-10 [100,105]. While this immune dampening is necessary for host survival, it is also likely one of the main reasons the parasite is able to persist into chronic phases [100]. Within the chronic phase, parasites are commonly undetectable in the blood and present at low levels in the tissue. It is known that continued CD8+ T cell activity is required to keep parasite levels in check in the chronic phase of infection [100], but how this parasite can persist at low levels long-term is not fully understood and is likely the result of a combination of factors such as antigenic diversity, dormancy/slowed replication, and parasite reservoirs in immune-privileged sites. The host immune response also plays a role in allowing parasite persistence, possibly through continued cytokine-dependent dampening of effector functions and T-cell exhaustion [100]. 

To date, there have been no vaccines for Chagas disease which have progressed to human clinical trials. Preclinical efforts have focused on both prophylactic and therapeutic vaccination, primarily using subunit vaccines combined with Th1 skewing adjuvants (See Table 2 for a non-exhaustive list of common adjuvants) [106]. Therefore, AdV vectors are a logical next step given their natural adjuvating and Th1 response driving abilities [31]. Research groups from Brazilian universities, in collaboration with the University of Massachusetts, Worchester, have published a significant body of work investigating AdV-vectored vaccines for Chagas disease in murine models. They have developed HAdV-5 vectors expressing *T. cruzi* amastigote surface protein-2 (ASP2) and/or trans-sialidase (TS) [107,108,109,110,111,112,113,114,115,116,117,118,119,120]. It was found that immunization with both AdV-ASP2 and AdV-TS vaccines together provided 100% survival against *T. cruzi* Y strain in BALB/c and C57BL/6 mice when challenged four weeks post-vaccination. All non-vaccinated BALB/c mice died within 20 days post-challenge, and 50% of non-vaccinated C57BL/6 mice died within 25 days [107]. As C57BL/6 mice preferentially develop Th1 immune responses [121], partial survival in the non-vaccinated group provides further evidence of the contributions of Th1 responses to Chagas disease protection. Based on these promising results, the authors conducted many studies with various modifications. In one instance, the use of an influenza vector prime/AdV boost, both expressing ASP2, was investigated. In C57BL/6 mice, only one AdV vaccine dose was needed to achieve 100% survival following the lethal challenge. In a more susceptible mouse model, influenza prime/AdV boost provided the greatest protection, significantly more than mice receiving either vaccine alone or homologous AdV prime/boost. The authors demonstrated that protection conferred by the influenza vector was epitope-specific as an influenza vector vaccine expressing the medial portion of ASP2 (ASP2-M) provided significantly greater protection than one expressing the carboxy region of ASP2. They also found that the use of heterologous vectors was important for generating protection. Mice primed with the influenza-ASP2-M and boosted with the AdV vector had a higher percentage of CD8+ T cells recognizing the immunodominant portion of ASP2 than mice that received the homologous AdV vaccine regimen encoding the same protein, which they hypothesize is because the influenza-ASP2-M vaccine drives a more specific immune response. The percentages of functional CD8+ T cells expressing IFN-γ, TNF-α, and/or CD107 were also higher in the group primed with influenza-ASP2-M [120]. 

To determine if the benefit of heterologous vaccination could be replicated using a different priming platform, the AdV vector vaccine was assessed when primed first by a DNA vaccine [108,109,110,111,112,115]. The vaccination scheme involved a DNA prime consisting of a combination of three plasmids, one each encoding ASP2, TS, and murine IL-12 as an adjuvant, followed by a mixture of AdV-ASP2 and AdV-TS boost. Protective efficacy was determined in a chronic challenge model evaluating cardiac complications. Significantly lower sinus arrhythmias and atrioventricular blocks were observed in mice vaccinated with the DNA prime/AdV boost over control mice at days 80, 150, and 240 days post-challenge with the Brazilian strain of *T. cruzi*. Cardiac abnormalities were also lowered in vaccinated mice following the challenge with the Colombian strain, but only the difference in sinus arrhythmias was significant. When this vaccine was tested in a therapeutic context, the prime and boost were administered 30 and 50 days after the challenge, respectively. There was no difference in parasitemia in vaccinated and control mice following priming; however, after boosting, there were significantly fewer sinus arrhythmias in vaccinated over control mice at days 90 and 120 following the challenge with the Brazilian strain. This difference was lost on days 150 and 180. Interestingly, the opposite pattern was observed when mice were challenged with the Colombian strain of *T. cruzi*, as DNA/AdV vaccination only provided significant protection at the later time points. Vaccination was not associated with any differences in atrioventricular blocks observed at any timepoint when animals were challenged with either strain [115]. These data suggest that although the IL-12 adjuvanted DNA prime/AdV boost demonstrated some protection from cardiac complications in a chronic challenge model, this vaccine was much less effective when used in a therapeutic context. Despite the lack of therapeutic protection observed, this research group attempted to change the parameters of the Chagas disease context by administering their therapeutic vaccine later in the course of the disease. Surprisingly, when delivered 120 days (prime) and 160 days (boost) post-infection, their vaccine strategy was able to ameliorate cardiac complications [116]. These contradictory results may indicate that the administration of this vaccine needs to occur later within chronic phase disease to demonstrate any benefit. When administered at 120 and 160 days post-infection, therapeutic vaccination reversed three important drivers of cardiac damage in chronic Chagas disease: connexin-43 (Cx43) disorganization, fibronectin (FN) deposition in the cardiac tissue, and increased CK-MB activity in the serum. Of note, therapeutic vaccination did not affect levels of the parasite in the heart, suggesting that the vaccine works by altering the host immunopathology rather than targeting the parasite itself [116]. This observation provides insight as to why delayed therapy after the onset of cardiac symptoms might be more beneficial. Two other reports demonstrate this vaccine’s efficacy when vaccination and challenge occur simultaneously [119,122]. Given that many infected individuals present as asymptomatic and their seropositivity will be unknown, the relevancy of these studies in a clinical setting is questionable; however, they can provide insight into the early immune responses to *T. cruzi* infection. Overall, while this vaccine has been investigated prophylactically and therapeutically in murine models, it should be further investigated in NHP and humans to determine its applicability to the field.

**Table 2 pharmaceuticals-16-00334-t002:** A non-exhaustive list of common adjuvants and their properties. Immune skewing effects are noted. However, the vaccine itself, as well as the route of administration, will have an effect on the skew as well. For a thorough summary of currently approved vaccine adjuvants, see [123].

Immune skew	Adjuvant	Composition	Sensor/Receptor	Use in Approved Vaccines	Refs.
Th1	IL12	Recombinant IL-12	IL12Rβ1 and IL12Rβ2		[124]
	Freund’s complete adjuvant	Mineral Oil-in-water + heat-killed mycobacteria	NOD2	Not for human use	[125]
	CpG	Synthetic oligodeoxynucleotides	TLR-9	Hepatitis B (Heplisav-B)	[126]
	MPL-A	Detoxified LPS	TLR-4	Component of AS04	[127]
	LPS	Lipopolysaccharide	TLR-4		[128]
	Flagellin	Flagellin	TLR-5		[129]
	AS01	Adjuvant system (MPL-A and QS-21)	TLR-4	Shingles (Shingrix), malaria (RTS,S/AS01)	[130]
Th2	Alum	Aluminum Salts	NLPR3, likely additional mechanisms	Diphtheria-Pertussis- Tetanus (Tdap), Hepatitis A & B (Twinrix), etc.	[131]
	Freund’s incomplete	Mineral Oil-in-water	Unknown	Human use discontinued in the 1950s	[125]
	Various helminth antigens		Various		[29,132]
Mixed	MF59	Oil-in-water	Unknown	Influenza (Fluad Quadrivalent)	[133]
	AS04	Adjuvant system (MPL-A and Alum)	TLR-4	HPV (Cervarix)	[127]
	AS03	Adjuvant system (α- tocopherol and squalene)	Unknown	Influenza (Pandemrix, Q-pan)	[134]
	Matrix M	Saponins	Unknown	COVID-19 (Nuvaxovid)	[135]
Mucosal	Cholera toxin B	Non-toxic subunit B of Cholera toxin	GM1	Cholera (Dukoral)	[136]
	dmLT	Modified heat-labile enterotoxin	GM1		[137]

Abbreviations used: MPL-A = Monophosphoryl lipid A, LPS = lipopolysaccharide, GM1 = monosialotetrahexosylganglioside, HPV = human papilloma virus.

Most AdV-vectored vaccines involve incorporation of transgenes encoding exogenous antigens into the AdV genome, resulting in secretion of said antigens following transcription and translation in the infected host cell. However, some researchers have utilized novel “capsid integration” strategies, designing AdV vectors that incorporate exogenous antigens into their viral capsid proteins, including multiple vaccines developed by a group at the University of Alabama at Birmingham [138,139,140]. One of their HAdV-5 vectors incorporated *T. cruzi* glycoprotein 83 (gp83) within the hypervariable region 1 of the HAdV-5 hexon protein. Mice were then vaccinated in a homologous prime/boost regimen. *T. cruzi*-specific serum IgG was significantly increased in vaccinated mice post-prime and post-boost, compared to controls. After the challenge with a lethal dose of *T. cruzi*, 60% of vaccinated mice remained alive after 27 days, while all control mice died by day 18. Parasitemia was also significantly lower in vaccinated mice after day 14 [138]. Based on these results, a second paper was published using the same vaccine in combination with another modified HAdV-5 virus containing a fragment of the ASP2 antigen (ASP2-M) incorporated into the minor capsid protein pIX. The authors first vaccinated mice with the AdV-pIX-ASP2 vaccine alone and found that PBMCs (peripheral blood mononuclear cells) from vaccinated mice contained significantly more IFN-γ producing cells (measured by ELIspot) than control mice. Using intracellular staining on splenocytes, they then found that there was a significantly higher percentage of CD8+ T cells producing CD107, TNF-α, or IFN-γ (and dual CD107/TNF-α or CD107/IFN-γ producing cells) in vaccinated mice over controls. After vaccination with either the AdV-gp63 or AdV-pIX-ASP2 vaccine alone or both together, animals were challenged with a lethal dose of the Tulahuen strain of *T. cruzi*. The dual vaccine provided the greatest decrease in parasitemia (80% lower than controls) and the greatest survival rate to day 55 (~70%), demonstrating significant protection [140]. Given that the transgene approach and capsid incorporation approach can result in subtly different immune responses, using both together likely broadens and enhances the overall immune response. In addition to their work with HAdV-5, this group also published a paper using a less seroprevalent human serotype of AdV, HAdV-48 [139]. They compared immune responses to HAdV-5 and HAdV-48 vaccines expressing a fragment of ASP2 (ASP2-C) or gp83 either expressed as a transgene or incorporated into the pIX protein. Both the HAdV-48-ASP2-C transgene and HAdV-48-ASP2-C pIX vaccines were able to induce a higher percentage of CD107/TNF-α or CD107/IFN-γ double positive CD4+ and CD8+ mouse splenocytes in comparison to the HAdV-5-ASP2 transgene vaccine. Additionally, HAdV-48-gp83 pIX and HAdV-48-ASP-C pIX vaccines induced higher serum IgG levels than the corresponding HAdV-5 vaccines [139]. This paper provides proof of principle for pIX-modified AdV vectors as promising *T. cruzi* vaccines. Moving forward, their protective efficacy will need to be tested in challenge models. The combined data described here present compelling evidence for the capsid integration strategy, in addition to the use of HAdV-48 as an alternative AdV serotype.

## 6. Adenovirus-Vectored Vaccines against Schistosomiasis

Schistosomiasis is caused by blood-dwelling trematode worms of the *Schistosoma* genus, most commonly *S. mansoni, S. japonicum,* and *S. haematobium* [141]. An estimated 240 million individuals are affected by schistosomiasis, and more than 700 million are at risk of infection [142]. Individuals are infected through contact with freshwater containing *Schistosoma* larvae (cercariae) shed by the parasite’s intermediate host, freshwater snail [142]. Schistosomiasis is considered the second most devastating parasitic disease after malaria in terms of worldwide impact [143]. Infection occurs in tropical and subtropical areas, with the greatest burden in Sub-Saharan Africa [142]. The three main species have distinct geographical distributions: *S. mansoni* occurs in Africa, the Middle East, the Caribbean, and parts of Latin America; *S. japonicum* occurs in China, Indonesia, and the Philippines; and *S. haematobium* occurs in Africa, the Middle East, and Corsica (France) [142]. There are two types of schistosomiases: intestinal and urogenital, classified by the species of *Schistosoma* and locus of adult worms within the mammalian host. Intestinal schistosomiasis occurs when parasites reside in the veins surrounding the intestines, and urogenital schistosomiasis occurs when they reside in the veins surrounding the urinary tract. Intestinal schistosomiasis is caused by *S. mansoni, S. japonicum*, and other less prominent species, while urogenital schistosomiasis is caused by *S. haematobium* [142]. While acute schistosomiasis can occur, in endemic areas the disease is typically chronic. Pathology develops in response to granuloma formation and fibrosis surrounding eggs deposited into the tissue, typically within the liver for intestinal schistosomiasis or in the bladder for urogenital schistosomiasis [141,144]. Chronic pathology is associated with high levels of morbidity; schistosomiasis causes an estimated 1.9 million disability-adjusted life-years (DALYs) [145] and estimates of between 24,000 and 200,000 annual deaths [142]. Infection is treated with praziquantel, often through mass drug administration (MDA) efforts. While MDA has had a positive impact on disease burden, it remains insufficient, especially in high-prevalence locations [146]. 

The immune response to *Schistosoma* infection is complex and multifaceted. Over the course of infection, the human host is exposed to four separate life stages of the parasite: two larval stages, cercariae and schistosomula; adult worms; and eggs. The different life stages present hundreds to thousands of shared and unique antigens, stimulating a wide range of humoral and cell-mediated responses [147,148,149]. Immediately upon entry, cercariae penetrating the skin upregulate regulatory immune responses to evade host defenses [147,150]. As cercariae transition to schistosomula, which passage through the lungs and circulatory networks, Th1 responses dominate and are maintained until the adult worm stage (for an overview of T helper subsets, see Table 3). Upon oviposition, there is a rapid switch to Th2 responses [147,150]. Granuloma formulation around eggs deposited in the tissue, and subsequent pathology, is driven by this Th2-type inflammatory response, activated largely by soluble egg antigens present on and secreted by *Schistosoma* eggs [151]. As the infection continues into chronic stages, the immune response shifts again in favor of Treg immunity, protecting both the parasite and the host [152]. The “Happy Valley” hypothesis, first proposed by R.A. Wilson, theorizes that if Th1 and Th2 responses were to be visualized on a linear continuum, schistosomes would thrive in the middle, or the valley floor, where there is a perfectly balanced Th1/Th2 response. In contrast, when the immune response is skewed in either direction, the sides of the valley, a stronger anti-parasite effect is achieved [153]. That said, mice with highly polarized Th1 or Th2 responses (IL-10/IL-4 and IL-10/IL-12 knockouts, respectively) have demonstrated distinct lethal immunopathology in response to *Schistosoma* infection [154]. Thus, neither a broadly balanced Th1/Th2 nor a completely polarized response is fully protective; a targeted response that maintains characteristics of both Th1 and Th2 responses but avoids the “Happy Valley” has been described as ideal for host protection [29]. Natural resistance to schistosomiasis in individuals living in endemic areas seems to develop, similarly to malaria, over the course of many years of repeated infection. This protection is thought to be induced when adult worms die and release immunogenic or “cryptic” antigens, promoting the production of cross-protective responses which are active against larval stages. In humans, IgE appears to be a key mediator of protection, although an IgE-based vaccine is unlikely to be feasible given its association with hypersensitivity reactions [150]. 

Efforts to develop a schistosomiasis vaccine began in the 1970s using radiation-attenuated cercariae (the infective stage of *Schistosoma* sp.) [156,157]. While these vaccines showed very significant protection in animal models, they proved impossible to successfully translate and scale-up for human use [144,158,159]. Perhaps the most important finding from these early experiments, as well as later trials, was that protection is best achieved with a combination of targeted Th1 and Th2 responses [29,158,159,160]. Most modern schistosomiasis vaccines are based on subunit approaches using a variety of immunogenic *Schistosoma* proteins [158].

To date, three AdV-vectored schistosomiasis vaccines have been developed, two against *S. japonicum* [41,42,43,44] and one against *S. mansoni* [45]. The first *S. japonicum* vaccine, developed by Dai et al., has thus far been studied in mice with the goal of creating a veterinary vaccine [41,42,43]. Unlike *S. mansoni* and *S. haematobium, S. japonicum* is primarily a zoonotic disease. While many animals can host the parasite, transmission is thought to be driven primarily through bovines, especially water buffalo [161]. The vaccine consists of an HAdV-5 vector expressing *S. japonicum* triosephosphate isomerase (SjTPI), a glycolytic pathway enzyme expressed at all stages of the *S. japonicum* life cycle [41,42,43,162]. Dai et al. first tested this vaccine using three routes of administration: intramuscular (IM), subcutaneous (SC), and oral immunization (PO). IM vaccination produced the highest titers and avidity of SjTPI-specific IgG and provided the greatest protection from challenge (50% reduction in worm/egg burden). Vaccination delivered IM caused a Th1 skewed response, while SC delivery caused a Th2 skew. Oral administration of AdVs has been tested before, including the live-attenuated AdV vaccines used by the United States military for many years [163]. However, in this study, PO administration induced little immune response and protection [44]. The authors posit that the AdV vector may be unable to survive the pH of the digestive tract, although the success of other PO-delivered vaccines using AdV vectors questions this hypothesis. Continued work on the AdSjTPI vaccine analyzed heterologous prime-boost regimens with the AdV-vectored vaccine delivered IM as the prime and protein (SjTPI) in Freund’s adjuvant-delivered SC as the boost. The heterologous AdV and protein vaccines afforded the broadest immune response, with significant Th2, Th17, and Th1 cytokine and antibody responses in comparison to either the AdV or protein alone. The heterologous vaccine regimen also provided the greatest protection, demonstrating an approximately 70% reduction in worm and egg burdens [42]. The authors also investigated various other heterologous prime-boost regimens using combinations of DNA, AdV, and protein, but their findings still supported the AdV prime/protein boost as the most efficacious option [43].

A second *S. japonicum* AdV-vectored vaccine used HAdV-5 as the vector and *S. japonicum* inhibitor apoptosis protein (SjIAP) as the expressed antigen. In mice trials, the authors saw an increase in IFN-γ and IL-2 production from isolated splenocytes following vaccination, suggesting a Th1 skewing, but a high IgG1 to IgG2a ratio, suggesting the Th2 skewing of humoral responses. The combination of the Th1-inducing AdV and Th2-inducing *Schistosoma* antigen is likely responsible for this dual Th1/Th2 reaction. In a murine challenge model, the vaccine only provided a 32% and 38% reduction in egg and worm burden, respectively [44], and further optimization has yet to be published.

The most recently developed AdV-vectored schistosomiasis vaccine targets the species *S. mansoni* [45], the most widespread cause of schistosomiasis globally [142]. This vaccine also uses a HAdV-5 vector. The antigen expressed, *S. mansoni* Cathepsin B (SmCB), is a cystine protease essential to parasite digestion of blood molecules and, thus, survival and development [45]. This vaccine has been tested in mice using a heterologous AdV prime/protein boost regimen with IM delivery. The heterologous vaccination provided significant protection against challenge, achieving approximately 70% reduction in worm and egg burdens. IgG responses showed a Th1 skewing effect of the vaccine. However, IgG1 levels were comparable to protein-alone vaccination, suggesting the maintenance of antigen-specific Th2 responses like was seen with the SjIAP AdV vaccine. Th1 cytokines, most notably IFN-γ, were also observed [45].

While no AdV-vectored vaccines for schistosomiasis have advanced to primate or human trials, results in mice for vaccines against both *S. japonicum* and *S. mansoni* have shown promising pre-clinical results. As was first demonstrated with irradiated cercarial vaccines, both Th1 and Th2 immune responses appear to be required for protection against challenge [41,42,43,45,160]. It remains to be seen whether these AdV-vectored vaccines will translate to their desired populations (humans, bovines), though continued research is certainly warranted. Notably, no AdV-vectored schistosomiasis vaccine has been developed to target *S. haematobium*; however, as SmCB shares an 84% homology with its *S. haematobium* counterpart, there is theoretical evidence for cross-protection conferred by AdV-SmCB. Cross-protection is especially important as most regions are co-endemic for *S. mansoni* and *S. haematobium*. Another consideration is the therapeutic potential of candidate vaccines, given many individuals in target locations may already be infected at the time of vaccination. Additional *Schistosoma* antigens have been tested in other vaccine efforts and may be amenable to incorporation into AdV vectors for future development [164].

## 7. Adenovirus-Vectored Vaccines against Leishmaniasis

Leishmaniases are diseases caused by protozoan parasites of the *Leishmania* genus. There are three major types of leishmaniasis: visceral leishmaniasis (VL), cutaneous leishmaniasis (CL), and mucocutaneous leishmaniasis (MCL). VL is the most dangerous form of the disease and has a 95% case fatality rate if untreated. CL is less severe, affecting only the skin, but is more common than VL. MCL is the rarest form of the disease and causes the destruction of mucous membranes, particularly in the nose and mouth [165]. A fourth clinical manifestation, post-kala-azar dermal leishmaniasis (PKDL), occurs in individuals cured of VL and is characterized by a skin rash without visceral symptoms [166]. The WHO identified 98 countries as endemic for leishmaniasis in 2021. However, the greatest burden is in South America, East Africa, and West and Southeast Asia [167]. VL occurs primarily in India, East Africa, and Brazil; CL occurs primarily in the Americas, the Middle East, central Asia, and the Mediterranean basin; and MCL occurs primarily in Bolivia, Brazil, Ethiopia, and Peru [165]. Up to 1 million cases of leishmaniasis are estimated to occur each year, and only a small percentage of those are reported to the WHO [165]. There are more than 50 known species of *Leishmania*, of which at least 20 infect humans [168]. The type of leishmaniasis is determined by the species causing the infection: VL is caused by *Leishmania donovani* and *Leishmania infantum,* while CL is caused by many species, including *Leishmania major, Leishmania mexicana,* and *Leishmania amazonensis* [168]. Transmission of *Leishmania* occurs through the bite of infected female sandflies of the genera *Phlebotomus* (Old World) and *Lutzomyia* (New World) [168].

Immune responses to *Leishmania* infection are well characterized. In broad terms, Th1 responses are protective, while Th2 responses allow parasite survival and proliferation. This Th1/Th2 dichotomy model is based on murine models of *L. major*, especially in BALB/c mice, where a Th2 dominant response leads to failure to control the infection [169,170,171,172,173]. While this model has been the standard since the 1970s, more recent research suggests the immune response to *Leishmania* may not be quite as simple, especially in VL [171,172,174]. The full picture of *Leishmania* immunity is complex and differs based on the infecting strain and host characteristics [174,175]. Whether there is a complete Th1/Th2 dichotomy or not, some correlates of protection for leishmaniasis are widely agreed upon. IFN-γ has been consistently identified as protective and necessary for disease control in human, mouse, canine, and NHP models of VL [175]. To that end, CD8+ T cell-mediated immunity is key to resistance to disease through the production of IFN-γ, killing infected cells by the Fas/FasL pathway, and even recruiting cells for protective granuloma formation [176,177]. Innate immune responses are also important in *Leishmania* infection, especially given the parasite replicates within macrophages and can infect other innate immune cells [172,178].The production of ROS and NO by infected and non-infected innate immune cells is also generally thought to be important in parasite killing [172]. The importance of NO and ROS production is demonstrated by the fact that iNOS-deficient mice are extremely susceptible to *Leishmania* infection [179], and infected monocytes with inhibited ROS pathways are similarly unable to kill the parasite [180]. 

While no vaccine has been approved for human leishmaniasis, there are many reasons why researchers believe one is possible. Since ancient times, various cultures have used Leishmanization as a form of inoculation against CL. Leishmanization refers to the practice of removing infectious material from a lesion on a CL patient and transferring that material to a healthy person, typically a child. More modern approaches have used cultured live *L. major* to achieve the same effect. This inoculation results in a typical CL lesion, which is usually self-healing, and provides the recipient protection from further infection. The practice has been used and studied widely up to the twenty and twenty-first centuries, notably in the former USSR, Iran, and Israel. Leishmanization provides evidence that immunity can be achieved, but the practice is generally considered too risky for use on a broad scale. No similar practice has been developed for VL [3], although CL Leishmanization has shown cross-protection from both diseases [181,182]. Current *Leishmania* vaccine development efforts can be classified into three generations. First-generation vaccines include live-attenuated parasites, whole-killed parasites, and fractionated antigens. Second-generation vaccines are recombinant proteins, often adjuvanted. Third-generation vaccines include DNA vaccines and vectored vaccines, including AdV vectors [183].

Two AdV-vectored leishmaniasis vaccines have been developed in the past ten years [184,185,186]. The first targets VL using a HAdV-5 vector expressing *L. donovani* amastigote specific antigen A2 (AdV-A2) and was tested in NHP (*Macaca mulatta*) [184]. Recombinant A2 protein is the basis of the Leish-Tec^®^ vaccine, one of the currently approved *Leishmania* vaccines for canines [184,187]. In their macaque study of AdV-A2, the authors investigated a variety of prime-boosting schedules using DNA, AdV-A2, and recombinant A2 protein with adjuvant (recombinant human interleukin-12 adsorbed in alum (rhIL-12/alum)) [184]. They found that both A2 and soluble *Leishmania* antigen (SLA) specific IgG was increased in serum post-boost from macaques vaccinated with AdV-A2 prime/protein boost but not in macaques vaccinated with DNA prime/AdV-A2 boost. IgG levels in both protein alone and AdV-A2/protein groups decreased after the challenge with *L. infantum.* All macaques developed some degree of infection, which the authors categorize as either “sub-patent” (low parasitism and asymptomatic), “asymptomatic” (patent parasitism), or “symptomatic” (patent parasitism and symptomatic). At 6 weeks post-infection, 100% of macaques vaccinated with the AdV-A2/protein regimen were asymptomatic, while 73% of control macaques were symptomatic. At 24 weeks, 100% of AdV-A2/protein vaccinated macaques had sub-patent infections, and 61% of control macaques remained symptomatic. Lower degrees of protection were observed in the other vaccine regimens. The authors also looked at granuloma formation and resolution in the liver. All macaques had granulomas 6 weeks post-infection, indicating all were infected, but at 24 weeks, macaques vaccinated with AdV-A2/protein showed complete granuloma resolution, while control macaques did not. Macaques vaccinated with the other regimens showed varying degrees of lesser resolution. This study presents evidence that AdV-A2 prime/protein boost provides a significant degree of protection from clinical disease, but not sterile immunity, in macaques [184]. As a future direction, the protective role of the AdV-vectored vaccine alone could be determined by directly comparing the AdV/protein heterologous regimen tested here to AdV/AdV and protein/protein homologous regimens. The authors did not investigate many of the typical correlates of immunity in *Leishmania* infection, such as IFN-γ production, and this vaccine has not yet been tested in humans.

The second AdV-vectored vaccine for leishmaniasis uses a ChAd63 vector expressing a fusion protein (KH) comprised of synthetic *L. donovani* antigens kinetoplastid membrane protein-11 (KMP-11) and hydrophilic acylated surface protein B (HASPB). HASPB contains a polymorphic repeat region, so the synthetic KH protein was designed to mimic that diversity, potentially allowing for cross-protection against different species of *Leishmania*. This vaccine was designed as a therapeutic vaccine for VL/PKDL and has been tested in mice [188], healthy human volunteers [185], and patients with persistent PKDL [186]. Most recently, this vaccine was tested for safety and immunogenicity in 23 Sudanese adults and adolescents with persistent PKDL (defined as lasting longer than 6 months). Vaccinated adult patients were followed for 90 days, and adolescents were followed for 120 days. During follow-up, 47.8% of patients had up to 25% clinical improvement, and 21.7% had 40–60% clinical improvement; 30.7% of patients did not require chemotherapy to resolve PKDL lesions. The authors then performed whole blood transcriptomic analysis to determine predictors of PKDL clearance. They found 11 gene modules with the highest predictive value, two of which were differentially expressed in resolving versus non-resolving patients: one containing genes encoding lysosomal/endosomal proteins and one containing genes enriched in monocytes. The authors also identified differentially expressed genes as early as 1 day post-vaccination, and they found enrichment in many innate immune response-related genes, including IFN type I and II signaling following vaccination. Finally, they also measured IFN-γ producing PBMCs by ELIspot and found an increase in the percentage of IFN-γ-producing cells post-vaccination compared to pre-vaccination in most subjects [186]. Despite these positive data, the small sample size in this study and the lack of a placebo arm makes it difficult to evaluate the effects of the vaccine on increased healing and the ability to self-cure. The authors note that PKDL in Sudan is particularly heterogenous and poorly understood, complicating the responses within subjects [186]. However, the presence of self-curing individuals in this trial is promising and warrants further study. A phase IIb randomized placebo-controlled trial, also in Sudan, began in 2019 and is currently ongoing (NCT03969134).

## 8. Adenovirus-Vectored Vaccines against Toxoplasmosis

Toxoplasmosis is a disease caused by the protozoan parasite *Toxoplasma gondii*. This parasite is geographically widespread, infecting up to a third of the world’s population with varying levels of seroprevalence around the globe [189,190,191]. Felids are the only definitive host for *T. gondii*; however, any warm-blooded animal can act as an intermediate host, and transmission can occur between both definitive and intermediate hosts without the other [192]. In humans, the infection can be food/waterborne, zoonotic, or congenitally transmitted from mother to child [193]. In rare cases, transmission can also occur through organ transplants or blood donations [193]. Despite high levels of infection with *T. gondii*, actual clinical disease is rare and generally confined to cases of immunocompromised individuals and congenital transmission [189,191,192,193]. Immunocompetent infected individuals are almost always asymptomatic; only approximately 10% of infections in immunocompetent individuals cause a self-limiting mild illness [189]. Congenital toxoplasmosis occurs when the mother experiences primary infection during or directly before pregnancy and passes the infection to the fetus. The risk to the fetus decreases when infection occurs later in the pregnancy. In the congenital transmission of *T. gondii*, 75% of cases remain subclinical; however, especially if a mother is infected within the first trimester, spontaneous abortion, prematurity, and events of stillbirth can occur. In surviving infants, clinical manifestations are varied, but the “classical triad” of congenital toxoplasmosis consists of chorioretinitis, intracranial calcifications, and hydrocephalus [194]. The second major risk group for clinical toxoplasmosis is immunocompromised individuals. Like in congenital cases, central nervous system (CNS) involvement is typical in immunocompromised hosts. Toxoplasmic encephalitis (TE) is the most common manifestation, but disseminated toxoplasmosis can also occur [192,195]. TE is characterized by a variety of symptoms, including mental status change, seizures, motor declines, sensory deficiencies, and neuropsychiatric conditions [189,195]. Although cases of severe disease are almost always limited to congenital and immunocompromised cases, a significant body of literature has identified correlations between *T. gondii* infection and a wide variety of conditions, especially neuropsychiatric diseases such as schizophrenia and psychosis [190,196]. A recently published analysis of ~14,000 individuals over ~22 years found a significantly higher all-cause mortality rate in seropositive versus seronegative individuals, indicating that there may be more to the story than we currently understand [197]. 

*T. gondii* is an obligate intracellular pathogen and can infect any nucleated cell, both through active penetration and phagocytosis [198]. Upon infection, a strong innate immune response is elicited, but the exact mechanisms of *T. gondii* sensing in humans have not been fully elucidated. In mice, TLR-11 and TLR-12 sense *T. gondii* profilin, leading to MyD88-dependent signaling and the production of IL-12. Humans do not have functional TLR-11 or 12 but still produce IL-12 in response to *T. gondii* infection [199,200,201,202]. The production of IL-12 drives the proliferation of NK cells, CD4+ T cells, and CD8+ T cells, all of which are significant producers of IFN-γ [203]. During acute infection, IFN-γ is the key effector. Production of CCL2 has also been demonstrated to be vital in early *T. gondii* responses in humans [201]. Although immunocompetent individuals mount a strong response to *T. gondii* early in infection, this is not sufficient to fully clear the parasite, which enters into its chronic cyst phase [199,200,204]. During the chronic phase, perforin-mediated, instead of IFN-γ mediated, CD8+ CTL activity is necessary to destroy cysts [205], and exhaustion of CD8+ and CD4+ T cells late in chronic infection can lead to reactivation and TE [206,207]. 

Toxoplasmosis vaccine research has been ongoing for decades [208,209], and although there are none yet approved for humans, a live attenuated vaccine, ToxoVax^®^, is licensed for the prevention of congenital toxoplasmosis in sheep [210]. Concerns about safety and shelf life have prevented this vaccine, and other live attenuated toxoplasmosis vaccines, from being tested in humans [208,209,211], providing a preference for the development of toxoplasmosis subunit vaccines. Multiple AdV-vectored vaccines against *T. gondii* have been researched and are thought to be excellent options for toxoplasmosis protection, given they strongly induce CD8+ T cell responses.

The Brazilian and American groups working on AdV-vectored *T. cruzi* vaccines have also developed toxoplasmosis vaccines using a HAdV-5 vector expressing surface antigens of *T. gondii* [212,213,214,215]. Most recently, they tested a heterologous prime-boost vaccination regimen with HAdV-5 (AdV-SAG1) and MVA (MVA-SAG1) expressing *T. gondii* surface antigen SAG1 in a murine challenge model, which they compared to a homologous AdV-SAG1 prime/boost protocol [215]. The authors found that the immune response was very similar between the two regimens. Although the authors did not show total IgG responses, they did find that both the heterologous and homologous protocols elevated IgG2c levels in the serum significantly over controls. IgG1 was also higher in vaccinated animals, but not significantly. They also found percentages of TNF-α and IFN-γ producing CD4+ and CD8+ T cells isolated from splenocytes were significantly higher in vaccinated over control mice. While the heterologous regimen consistently had higher percentages of cytokine-producing CD4+ and CD8+ T cells than the homologous protocol, the only significant difference observed between the two regimens was in the IFN-γ producing CD8+ T cell population. Notably, there was a significant increase in the percent survival following the *T. gondii* challenge for the heterologous protocol over the homologous, both being higher than the controls. This heterologous approach is similar to the ChAd63/MVA malaria vaccines, which again highlights the broad applicability of prime/boost using different viral vectors as a protective strategy.

Two vaccines using canine AdV 2 (CAV-2) expressing *T. gondii* virulence factors ROP18 and ROP16 have also been developed and tested in mice [216,217]. The CAV-2-ROP18 vaccine induced significant IgG titers over controls from 2–6 weeks post-vaccination, and the IgG2a to IgG1 ratio in vaccinated mice was 2.69 [216]. The IgG subclass results suggest a significant Th1 skewing, which should be protective against toxoplasmosis. A significantly larger production of splenocyte-isolated IL-2 and IFN-γ, as well as a larger proportion of CD4+ and CD8+ T cells expressing TNF-α and IFN-γ, were observed in vaccinated over control mice. After intraperitoneal (IP) injection of 1 × 10^3^ tachyzoites of the *T. gondii* RH strain, all control mice died after 7 days, while 40% of vaccinated mice survived to day 60. The authors also used an intragastric challenge, which mimics human infection and creates a latent chronic infection, allowing for the analysis of protection from brain cysts. Using this model, they observed a 57.3% reduction in brain cysts in vaccinated over control mice [216]. The CAV-2-ROP16 vaccine had a very similar immunogenic profile to the ROP18 vaccine; however, it only provided 25% protection from the IP challenge [216]. The endpoint of the ROP16 study was 80 days instead of 60, though the same percentage of animals remained alive at day 60 [217]. These vaccines, especially CAV-2-ROP18, show promise for further development given the significant protection even after 60 days and the induction of strong cellular and humoral immune responses. Intragastric challenge more closely resembles natural infection in humans and may be a more predictive model of actual vaccine efficacy in the field. It will also be important to test this vaccine in models which better represent the main risk groups for human toxoplasmosis, namely immunocompromised and pregnant individuals. 

Another anti-*T. gondii* vaccine was developed using a HAdV-5 vector and expressing ubiquitin conjugated multistage antigens (UMAS). UMAS is comprised of portions of *T. gondii* SAG3, ROP18, MIC6, GRA7, MAG1, BAG1 and SPA antigens which together represent all three stages of the *T. gondii* life cycle (tachyzoites, bradyzoites, and sporozoites). Mice were vaccinated with either a DNA vaccine encoding UMAS or an AdV vector expressing UMAS. The DNA vaccine alone conferred a strong serum IgG response but lower levels of IFN-γ and IL-2 produced by splenocytes, whereas the AdV vaccine alone showed the inverse results. Thus, the authors combined the two in a heterologous prime/boost strategy. Interestingly, priming with the DNA vaccine and boosting with the AdV enhanced both the humoral and cell-mediated responses compared to either alone, but when the order of prime/boost administration was reversed, the same enhanced responses were not reflected. DNA prime/AdV boost has been proven to be an effective regimen in many diseases, including malaria and Chagas disease [118,218]. After assessing the immunogenicity of the various regimens, vaccinated mice were challenged with either a lethal or non-lethal strain of *T. gondii*. In the lethal challenge model, 67% of the DNA-prime/AdV-boost group survived for 30 days. Lower survival rates were seen with the other vaccine regimens, and all control mice died within 10 days. In the non-lethal model, the authors used brain cysts as a measure of protection and found a significant decrease in cysts in the DNA-prime/AdV-boost over the AdV-prime/DNA-boost or either vaccine alone [219]. This study presents promising results for the use of a UMAS-expressing DNA-prime/AdV-boost vaccine. It should be noted that the survival challenge study timeline is significantly shorter in this paper than those in the CAV-2-ROP16/18 studies, both of which had nearly 90% protection at day 30 [216,217]. The same group also tested their AdV-UMAS vaccine in mice administered by various routes: IM, SC, PO, intranasal (IN), and intravenous (IV). Serum IgG responses were generally consistent across vaccination routes, with IM inducing the highest titers and PO the lowest. The IgG2a response was greater than the IgG1 response in all groups, suggesting Th1 skewing of humoral responses. Serum IgA was significantly higher in mice vaccinated by mucosal routes (PO and IN) compared to IM, IV, and SC delivery, and IL-2 and IFN-γ production from splenocytes were highest in the IN group. The authors again challenged with lethal and non-lethal strains of *T. gondii* and used survival and brain cysts as the measures of protection, respectively. In the lethal challenge, IM, IN, and PO all maintained ~50% survival to day 28, while the IV and SC groups had a ~40% survival rate. In the non-lethal model, brain cysts were reduced in all groups compared to their controls, with the greatest reduction seen in IN and PO groups [220]. This work supports the investigation of mucosal AdV-vectored vaccination for *T. gondii* and potentially other parasites.

In recent years, there has been a shift toward the mucosal delivery of vaccines. Many infectious agents enter and establish through mucosal surfaces, including the lungs, gastrointestinal tract, and reproductive tract. To combat this, the body has specific mucosal immune responses that are robust but localized. Typical systemic vaccination does not elicit strong mucosal immunity, which is a major limitation within the context of mucosal pathogens [221]. All mucosal vaccines currently licensed for human use are live-attenuated or inactivated pathogens. The Dukoral cholera vaccine, however, does contain a subunit antigen, cholera toxin B subunit, in addition to the killed whole-cell *Vibrio cholerae* [221]. The success of this vaccine suggests that specific immunogenic antigens can be used as adjuvants with attenuated or killed pathogen-based vaccines or viral/bacterial vectors. Viral vectors can also more successfully deliver genetic material to mucosal surfaces than naked nucleic acids [221].

Multiple mono-, bi-, and trivalent AdV vaccines were developed to express combinations of *T. gondii* microneme protein 3 (MIC3), rhoptry protein 9 (ROP9), and surface antigen 2 (SAG2) [222]. In all vaccinated groups, *T. gondii*-specific serum IgG was significantly elevated over the controls. The trivalent vaccine had the highest IgG titers, followed by the SAG2-MIC3 and ROP9-MIC3 bivalent groups, while the SAG2-ROP9 group was lower than the monovalent MIC3 group. The ROP9 and SAG2 monovalent groups had the second lowest and lowest IgG levels, respectively. The authors looked at a variety of cytokines in the serum, most of which were significantly upregulated in vaccinated groups, notably IFN-γ and TNF-α, showing a strong Th1 response, as well as IL-10. IL-10 is important in the context of toxoplasmosis to prevent excessive inflammation in TE [199,204], but given IL-10 is an anti-inflammatory cytokine, it would not necessarily be considered protective in a vaccine model. Challenge was performed with “low” and “high” doses of a lethal strain, with the “high” dose being consistent with that used in the other studies presented in this review. Survival rates at 32 days post-infection were 40% for the bivalent AdV-MIC3-ROP9 vaccine, 30% for the trivalent and AdV-SAG2-MIC3 vaccines, 10% in the AdV-SAG2-ROP9, and 0% in the monovalent vaccines, although all groups did have significantly longer survival times than the non-vaccinated controls. Overall, the trivalent and bivalent vaccines were more immunogenic and protective than the monovalent vaccines. MIC3 seems to be important to the vaccines’ efficacy, given that the bivalent vaccine without this antigen consistently had lower responses and protection. The induction of IL-10 by this vaccine is interesting and has been observed with other *T. gondii* vaccines [223]. As IL-10 is not considered a classical correlate of protection for many pathogens, it often goes uninvestigated. Despite this, it is worth investigating whether IL-10 expression is common in other vaccines and whether vaccine-induced IL-10 could be deleterious or provide protection. 

*Neospora caninum,* the causative agent of neosporosis, is very closely related to *T. gondii* and was not identified as a separate species until 1988 [224]; as such, a cross-protective AdV vaccine was constructed to target both diseases [225]. Neosporosis mostly occurs in cattle and dogs, with canines being the definitive host [226]. Although no infections have been identified in humans [226], two macaques have been infected experimentally [227]. The toxoplasmosis-neosporosis vaccine consists of a HAdV-5 vector expressing a cross-reactive portion of the *N. caninum* apical membrane antigen 1 (AMA1) [225]. Post-vaccination, serum IgG, IFN-γ, and IL-4 were significantly increased in vaccinated mice over PBS controls. Vaccinated mice were challenged with *N. caninum* or *T. gondii*. The survival rate at 30 days post-challenge with *N. caninum* was 75% in vaccinated mice and 20% in control mice. At the same time point post-*T. gondii* challenge, survival was 45% in vaccinated and 0% in control mice [225]. The cross-protective abilities of this vaccine are very compelling and provide preliminary data that such a vaccine could work for other related species. AMA1 is also present in other Apicomplexan parasites such as *Plasmodium*, and many vaccines have been developed using the *Plasmodium* homolog of this antigen [218]. The potential for cross-protection between more Apicomplexan species using this antigen or others with significant homology could be further investigated and may be of interest to regions of co-endemicity.

## 9. Conclusions and Future Directions

The path toward parasite vaccine development has been arduous and is far from complete. While effective vaccines against viral and bacterial infections have existed for hundreds of years [4], no anti-parasite vaccine has been recommended for broad use before the RTS,S vaccine against *P. falciparum* malaria in 2021 [20]. Even then, the RTS,S vaccine has limited efficacy, especially compared to other licensed vaccines [23]. The need for better vaccines against malaria, as well as vaccines against other parasites, remains of paramount importance to global health and development. Viral vectors have emerged as a solution for pathogens historically left behind by traditional vaccine approaches, such as HIV, tuberculosis, and parasitic infections [13]. Many viral vectors exist, including poxviruses, lentiviruses, and AdVs, with AdV vectors being among the most widely used viral vectors in vaccine development, due to their favorable safety profile and a powerful ability to induce transgene specific immune responses, especially CD8+ T cell responses [8,9,31].

The majority of AdV-vectored vaccines against parasites have targeted protozoan parasites. CD8+ T cells are vital to control protozoan infections, as with other intracellular pathogens, making them especially good candidates for AdV-vectored vaccines [228]. In the past ten years, very significant advances have been made toward the development of AdV-vectored vaccines for malaria, leishmaniasis, Chagas disease, and toxoplasmosis. Vaccines for malaria using a heterologous prime/boost strategy with ChAd63 and MVA encoding ME-TRAP have advanced the furthest, completing phase IIb clinical trials [54,55,56,57,58,59,60,61,62,63,66,71]. A ChAd63 vaccine for leishmaniasis has also entered clinical trials, recently completing phase IIa trials as a therapeutic vaccine for patients with persistent PKDL [186]. AdV-vectored vaccines for Chagas disease and toxoplasmosis remain in preclinical testing; however, there are multiple promising vector/antigen combinations in the works.

Interestingly, AdV vaccines have also demonstrated protection against helminthic infections. Research into AdV-vectored vaccines against helminths has been mostly limited to vaccines against *Schistosoma* species, in particular *S. mansoni* and *S. japonicum*. Protection against schistosomiasis has long been known to require a delicately balanced Th1/Th2 response [29,158,159,160]. Due to the contributions of Th1 responses, schistosomiasis is a logical target for CD8+ T cell-inducing vaccines such as AdV vectors, although the mechanisms by which cell-mediated immunity targets large eukaryotic parasites are yet to be fully elucidated. While most helminth infections are characterized as purely Th2 driving, there is growing evidence for the role of Th1 immunity in protection against a number of helminths, including the lymphatic filariasis parasite *Brugia malayi* [29]. In addition to the schistosomiasis vaccines presented here, one veterinary vaccine against helminth *Taenia ovis* was developed over twenty years ago, though not pursued [229]. Similar to schistosomiasis, taeniasis appears to have an early Th1 response followed by a longer-term Th2 response [230,231]. Taken together, these vaccines highlight the potential for successful AdV vaccines against not only protozoa but helminth infections as well (see Table 4 for a summary of AdV vaccines against parasites).

Unsurprisingly, the major correlates of protection identified in AdV vaccines for parasites have been strong Th1-type responses, including high levels of IFN-γ, IL-12, and other proinflammatory cytokines, as well as IgG2-skewed humoral responses. These major immune responses, and the others discussed in this review, have been summarized in Figure 2, depicting their respective parasite targets.

For many parasites, the nuances of immune protection, modulation, and pathology are still not fully understood. Additionally, because natural immunity to parasites is often slow to develop, simply attempting to mimic those responses with vaccination is likely an ineffective approach. The development of effective anti-parasite vaccines will likely improve as the understanding of parasite immunity continues to deepen.

There are many important considerations in the development of AdV-vectored vaccines against parasites. First, there is the selection of antigen(s). Most parasites undergo multiple life stages within their human hosts, each with different dominant antigens. One approach to combat this diversity is the use of multiple antigens or epitopes within one vector construct. Examples discussed in this review include the *T. gondii* ubiquitin-conjugated multi-stage antigen segments [220], the *P. falciparum* TRAP antigen conjugated to a multi-epitope string [54,55,56,57,58,59,60,61,62,63,66,71], and the *T. gondii* trivalent AdV expressing three antigens of interest [222]. Another important consideration is the choice of vector. HAdV-5 vectors are the best characterized and are easy to manipulate; however, the high level of pre-existing immunity in humans is a concern [8,31,32,234]. Nonhuman AdVs have emerged as a potential solution, especially chimpanzee AdVs. ChAd63 has been used in the most successful parasite AdV-vectored vaccines to date for both malaria and leishmaniasis [54,55,56,57,58,59,60,61,62,63,66,71,185,186]. Other vectors, including rare human serotypes and canine AdVs, have also shown efficacy. Once antigens and vectors are selected, the last major decisions are vaccination schedules and modes of delivery, as all these variables can contribute to and augment varying immune responses. Heterologous prime boosting has been gaining traction as a more effective vaccination strategy for a number of pathogens. This method circumvents the problem of the development of neutralizing antibodies against the AdV vector, which might prevent the vector from effectively delivering the antigen and eliciting an immune response. Heterologous prime boosting can also increase both the strength and scope of the total immune response [235], which is especially important in parasitic diseases where correlates of protection are complex and multifaceted. The ChAd63/MVA vaccines for malaria are the best-described heterologous prime/boost method using AdV vectors; however, a number of other regimens are also very promising. In the context of schistosomiasis, where both Th1 and Th2 responses are needed for protection, AdV prime with protein boost is an excellent option [45]. This approach could be applied to other helminth infections, including lymphatic filariasis and taeniasis. The route of administration is also a key factor in the characteristics of an immune response. The AdV-UMAS vaccine for *T. gondii* showed considerably greater efficacy when administered mucosally versus systemically [220]; however, the AdSjTPI vaccine for *S. japonicum* had almost no protection when delivered orally [41]. It is worth investigating whether systemic or mucosal vaccination is generally more effective in the context of AdV-vectored parasite vaccines. There has been a notable shift towards mucosal vaccine development/testing for infectious diseases, and this change will likely filter into parasite vaccine development as well [221], as parasites have complex life-cycles which often occur adjacent to mucosal sites. A common theme emerging for parasite vaccines is the need for diversity in all facets: from antigen selection to delivery strategy [236]. The logic of prime/boost approaches can be applied to delivery methods as well, further increasing the options available to optimize vaccine success.

Parasitic infections have long been on the margins of vaccinology, both due to their complex immune interactions and general status as neglected tropical diseases [18]. Novel strategies, including AdV vectors, offer a promising avenue to finally provide protection to the billions at risk of parasitic diseases. Incredible progress has been made, but significantly more work is needed. AdV-vectored vaccines can, and should, be investigated for other parasites, including helminths. New antigens, vectors, regimens, and modes of delivery should be tested. Most importantly, there is a need for increased translational research to take these promising vaccine candidates from the bench to the bedside. To date, the only parasite targeting AdV-vectored vaccines to advance to clinical trials are against malaria and leishmaniasis. There is a very significant body of preclinical data using this vaccine strategy for schistosomiasis, Chagas disease, and toxoplasmosis, but none have begun trials in humans; in fact, the majority have only been tested in mice. To fully realize the potential of AdV-vectored vaccines for parasites, greater investment in translational research is needed across the globe.

## Figures and Tables

**Figure 1 pharmaceuticals-16-00334-f001:**
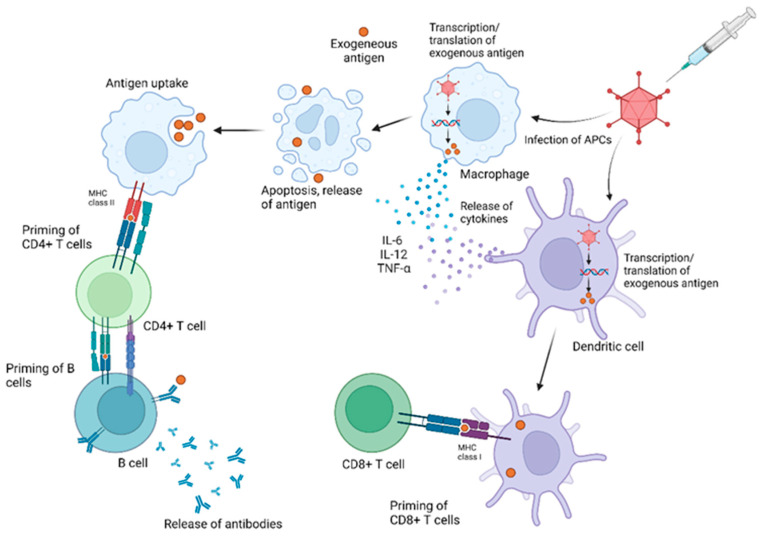
Immune responses elicited by adenovirus (AdV)-vectored vaccines. The AdV vector infects multiple cell types, including antigen-presenting cells (APCs) such as macrophages and dendritic cells (DCs). Once infected, these cells produce cytokines such as IL-6, IL-12, and TNF-α. Within DCs, transcription and translation of the antigen encoded in the AdV vector results in the presentation of the antigen on MHC class I to CD8+ T cells. Infected macrophages, and other cell types, which die by apoptosis or other means, release the antigen into the extracellular milieu, where the antigen can be taken up by other macrophages and presented on MHC class II to CD4+ T cells. CD4+ T cells, among other roles, prime B cells to produce antibodies against the antigen. Created with BioRender.com.

**Figure 2 pharmaceuticals-16-00334-f002:**
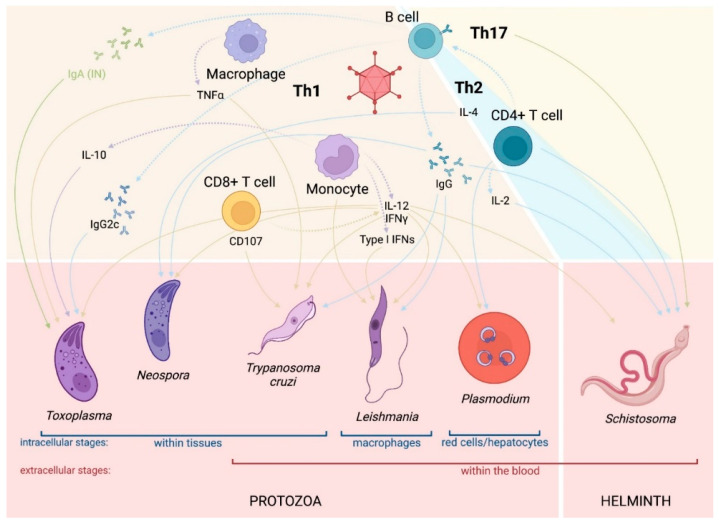
AdV vector immune effectors. The following is a graphical representation of the various immune effectors which have been shown by the papers herein to be activated by AdV vaccines. Included are hypothetical pathways (shown in dashed lines) contributing to cell-mediated and humoral immunity, which then target various protozoan and helminth parasites. Broadly, AdV vectors stimulate monocytes, macrophages, B cells, and CD4+ and CD8+ T cells within Th1-, Th2-, and Th-17 immunity (specific to *Schistosoma* AdV vaccines). These cells lead to the production of major immune effectors such as IL-12, IFN-γ, TNF-α, and antigen-specific IgG. Uniquely, AdV vaccines against *T. gondii* have shown associated production of IL-10, and when delivered intranasally (IN), they were also capable of eliciting antigen-specific IgA. Created with BioRender.com.

**Table 1 pharmaceuticals-16-00334-t001:** AdV-vectored vaccines in human clinical trials. This list is non-exhaustive.

Type of Pathogen or Condition	Disease	Vector	Phase	Status	ID Number
Virus	COVID-19	ChAdOx1	II/III	Active	NCT04400838
		HAdV-26	III	Active	NCT04505722
	Respiratory syncytial virus	HAdV-26	III	Recruiting	NCT04908683
	Norovirus	HAdV-5	IIb	Recruiting	NCT05212168
	HIV/AIDS	HAdV-26	III	Active	NCT03964415
	Influenza	HAdV-5	II	Completed, with results	NCT02918006
	Ebola	HAdV-26	III	Completed, with results	NCT02509494
Bacteria	Tuberculosis	ChAdOx1	I/IIa	Completed	NCT03681860
Parasites	*P. falciparum* Malaria	HAdV-35/HAdV-26	I/IIa	Completed	NCT01397227
		ChAd63	II	Completed	NCT01666925
	*P. vivax* Malaria	ChAd63	II	Completed	NCT04009096
	Leishmaniasis	ChAd63	IIb	Active	NCT03969134
Cancer	Advanced/metastatic solid tumors	HAdV-5	I/II	Active	NCT02285816
	Non-small cell lung cancer	ChAdOx1	I/II	Recruiting	NCT04908111
	Lynch syndrome cancer prevention	Gad20	Ib/II	Recruiting	NCT05078866
Drug dependence	Cocaine dependence	HAdV-5	I	Recruiting	NCT02455479

**Table 3 pharmaceuticals-16-00334-t003:** Characteristics of T helper (CD4+ T) cell subtypes and their associated immune responses [155].

T Helper Subset	Polarizing Cytokines	Master Transcription Factor	Key Effector Cytokines	General Targets	Related Cells/Effectors
Th1	IFN-γ IL-12	T-bet	IFN-γ	Intracellular pathogens	CD8+ T cells M1 macrophages Natural killer cells (NKs) IgG2
Th2	IL-4 IL-2	GATA-3	IL-4 IL-5 IL-13	Helminths/large extracellular pathogens	M2 macrophages Eosinophils IgE Goblet cells IgG1
Th17	IL-1β IL-6 IL-23	RORγt	IL-17A IL-17F IL-22	Extracellular bacteria and fungi	Nitric oxide Antimicrobial peptides Neutrophils
Treg	IL-2 TGF-β	Foxp3	IL-10 TGF-β IL-35	Regulation of the immune response	IL-2 consumption Negative signal transmission (CTLA-4, CD39 etc.) Perforin/granzyme B killing of antigen-presenting cells
Tfh	IL-6 IL-21	Bcl6	IL-21	Promotion of B cell maturation and antibody production	B cells CXCR5 Antibodies

**Table 4 pharmaceuticals-16-00334-t004:** Notable parasite AdV-vectored vaccines in pre-clinical and clinical trials (malaria excluded). This table is non-exhaustive. For summaries of AdV-vectored vaccines for malaria, see [232,233].

Disease	Parasite Speices	Antigen	Vector	Current Stage of Development	Refs.
Chagas disease	*T. cruzi*	ASP2 and/or TS	HAdV-5	Mice	[107,108,109,110,111,112,113,114,115,116,117,118,119,120,122]
		ASP2	HAdV-5	Mice	[139,140]
		gp83	HAdV-5	Mice	[138,139]
		ASP2	HAdV-48	Mice	[139]
		gp83	HAdV-48	Mice	
Schistosomiasis	*S. mansoni*	SmCB	HAdV-5	Mice	[45]
	*S. japonicum*	SjTPI	HAdV-5	Mice	[41,42,43]
		SjIAP	HAdV-5	Mice	[44]
Leishmaniasis and PKDL	unspecific	KMP-11 and HASBP	ChAd63	Clinical trials phase IIa	[185,186]
	*L. infantum*	A2	HAdV-5	NHP(Rhesus macaque)	[184]
Toxoplasmosis	*T. gondii*	TgMIC3	HAdV-5	Mice	[222]
		TgSAG2			
		TgROP9			
		UMAS	HAdV-5	Mice	[219,220]
		ROP16	CAV-2	Mice	[217]
		ROP18	CAV-2	Mice	[216]
		SAG1	HAdV-5	Mice	[215]
Toxoplasmosis/Neosporosis	*T. gondii* and *N. caninum*	NcAMA1	HAdV-5	Mice	[225]

## Data Availability

Not applicable.

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
