# Peer review of "Recent Advances in the Development of Adenovirus-Vectored Vaccines for Parasitic Infections"

_pharmaceuticals, 2023, doi:10.3390/ph16030334_

Round 1
Reviewer 1 Report
The manuscript is very interesting, well-written and informative. However, the following aspects need to be addressed before accepted for publication-
1. This manuscript mainly highlight the problems and prospects of vectored vaccines of parasites, therefore, I would like to suggest to reduce the descriptions regarding the pathology and impacts of each disease. In fact, these descriptions do not fit with the title of the articles as well.
2. In my point of view, reviews are written by experts, which is extremely helpful for the beginners. Therefore, I strongly suggest to give a table containing a list of adjuvants which help to induce Th1 or Th1 or Th1/Th2 biased immunity.
3. Please provide a table containing a list of cytokines/chemokines that are the indicators for Th1 or Th2 biased immunity.
4. Give a list of transcription factors along with their targets and particularly mention which transcription factors are associated for immunity against parasitic immunity.
Minor
Please do not start sentences with the short name of a parasite.
At line 76, after parasite put a ref, Anisuzzaman and Tsuji 2020 can be consulted.
At line 509, after S. haematobium put a ref, Labony SS et al 2022 (Pathogens) can be consulted.
Regarding scistosome biology Anisuzzaman et al 2021 and Frahm et al 2019 can be consulted.
Reviewer 2 Report
Review comments on pharmaceuticals-2168800
Dear Editor,
The manuscript by Koger-Pease et al., Recent advances in the development of adenovirus-vectored vaccines for parasitic infections reviews the efforts that have been made toward the development of vaccines to combat different parasitic infections, including malaria, schistosomiasis, and others. The authors present details and the rationale for the use of adenovirus as a vector for vaccine development and also indicate some of the challenges. The manuscript is well-structured and easy to understand.
I have carefully read through the manuscript and have these few comments to make to improve on understanding and add clarity.
1. Line 71, it will be a good thing for the authors to include selected vaccines against COVID-19 and include key details about these (efficacy, safety, etc.)
2. Concerning the structure, I think it important to mention the disease for each of the 5 diseases being reviewed in the paper. Data on disease burden is missing for some of the diseases. Also, it will be important to indicate information about the vectors for the diseases at beginning of each chapter that focuses on a particular disease. In addition, it will be important to indicate the geographical distribution of Schistosoma spp at the introduction of the chapter that focuses on schistosomiasis. The same thing goes for Leishmania spp
3. Lines 323-331, though R21 is not an adenovirus-vectored vaccine, I think it is important for the authors to provide basic information about this vaccine. This is particularly important because of the interesting data that is associated with the R21 vaccine.
4. The manuscript in some places has long sentences that could obscure meaning and reduce clarity. Also, a few spelling errors in some places (e.g., perforin-mediated instead of medicated, line 782-783, infectious agents and not infections, line 871). The authors should there carefully read again and make the necessary corrections.
5. In lines 378-380 where the authors talk about a delay in CD8+ responses to T. cruzi, it will important to indicate or suggest the possible reasons for this observation.
6. In discussing why heterologous prime boosting is gaining traction, the authors indicate that it circumvents the development of neutralizing antibodies. This statement seems to be unclear because in some cases, neutralizing antibodies are actually required for protection. More light should be thrown on this.
